# The prevalence of respectful maternity care during childbirth and its determinants in Ethiopia: A systematic review and meta-analysis

Aklilu Habte[1]◉*, Aiggan Tamene[1]◉, Demelash Woldeyohannes[1]◉, Fitsum Endale[1‡], Biruk Bogale[2‡], Addisalem Gizachew[1‡]

1 School of Public Health, College of Medicine and Health Sciences, Wachemo University, Hosanna, Ethiopia, 2 School of Public Health, College of Medicine and Health Sciences, Mizan-Tepi University, Mizan Aman, Ethiopia

◉ These authors contributed equally to this work.
‡ FE, BB and AG also contributed equally to this work.
* akliluhabte57@gmail.com

## Abstract

### Background

Respectful maternity care is the provision of woman-centered health care during childbirth that is friendly, abuse-free, timely, and discrimination-free. Although several epidemiological studies on the magnitude and determinants of Respectful maternity care in Ethiopia have been conducted, the results have been inconsistent and varied. This makes drawing equivocal conclusions and evidence at the national level harder. Hence, this systematic review and meta-analysis aimed at estimating the pooled prevalence of respectful maternity care and its determinants in Ethiopia.

### Methods

Studies conducted from 2013 to June 30, 2022, were searched by using PubMed, Google Scholar, Science Direct, Scopus, ProQuest, Web of Science, Cochrane Library, and Direct of Open Access Journals. Searching was carried out from May 15- June 30, 2022. In total, sixteen studies were considered in the final analysis. The data were extracted using Microsoft Excel and analyzed using STATA 16 software. The methodological quality of included studies was assessed by using Joanna Briggs Institute's critical appraisal checklist for prevalence studies. To estimate the pooled national prevalence of respectful maternity care, a random effect model with a DerSimonian Laird method was used. To assess the heterogeneity of the included studies, the Cochrane Q test statistics and $I^2$ tests were used. To detect the presence of publication bias, a funnel plot and Begg's and Egger's tests were used.

### Results

Sixteen studies were eligible for this systematic review and meta-analysis with a total of 6354 study participants. The overall pooled prevalence of respectful maternity care in

**Data Availability Statement:** All relevant data are within the paper and its Supporting information files.

**Funding:** The author(s) received no specific funding for this work.

**Competing interests:** The authors have declared that no competing interests exist.

**Abbreviations:** AOR, Adjusted Odds Ratio; CRC, Compassionate, Respectful and caring; FMOH, Federal Ministry of Health; HCPs, Health Care Providers; HSTP, Health Sector Transformation Plan; JBI, Joanna Briggs Institute; LMICs, Low and Middle-Income Countries; MNCH, Materna, neonatal and child health; PRISMA, Preferred Reporting Items for Systematic Reviews and Meta-Analysis; RMC, Respectful maternity care; SNNPR, South Nations and Nationalities People of the Region; WHO, World Health Organization.

Ethiopia was 48.44% (95% CI: 39.02–57.87). Receiving service by CRC-trained health care providers [AOR: 4.09, 95% CI: 1.73, 6.44], having ANC visits [AOR: 2.34, 95% CI: 1.62, 3.06], planning status of the pregnancy [AOR = 4.43, 95% CI: 2.74, 6.12], giving birth during the daytime [AOR: 2.61, 95% CI: 1.92, 3.31], and experiencing an obstetric complication [AOR: 0.46, 95% CI: 0.30, 0.61] were identified as determinants of RMC.

## Conclusion

As per this meta-analysis, the prevalence of respectful maternity care in Ethiopia was low. Managers in the health sector should give due emphasis to the provision of Compassionate, Respectful, and Care(CRC) training for healthcare providers, who work at maternity service delivery points. Stakeholders need to work to increase the uptake of prenatal care to improve client-provider relationships across a continuum of care. Human resource managers should assign an adequate number of health care providers to the night-shift duties to reduce the workload on obstetric providers.

## Introduction

Labor, in particular, delivery is a sensitive and vulnerable time in a woman's life [1]. Every woman has the right to woman-centered healthcare that is safe, effective, timely, respectful, and free of violence and discrimination during pregnancy, labor, and childbirth [2, 3]. Maternity care is a service that focused on improving maternal and newborn health outcomes during pregnancy, childbirth, and the postpartum period [4]. It includes monitoring the mother's and baby's well-being, health education, and assistance during childbirth [5].

Every year, around 140 million births occur worldwide, the vast majority of which are vaginal births with little difficulty for women and their newborns [6]. Pain, anxiety, threat, and exposure to the circumstance are the causes of women's vulnerability during labor and delivery [1]. Despite significant advances in maternal and child health, there is still a high rate of maternal and neonatal deaths globally [7]. Poor childbirth care contributes directly and indirectly to 82 percent of this problem [5]. Governments are striving to improve the quality of clinical care provided to women throughout pregnancy, and childbirth to achieve the global maternal mortality ratio target of 70 per 100,000 live births by 2030 [8, 9].

Following mounting evidence of mistreatment of women during pregnancy and childbirth around the world, the WHO declared the prevention and elimination of disrespect and abuse during childbirth by implementing the Respectful Maternity Care (RMC) initiative [10, 11]. RMC is one of the WHO's eight dimensions for quality maternal and newborn health care, and it refers to care that includes the right to access friendly, abuse-free, timely, and discrimination-free maternal health care, along with privacy, confidentiality, equality, informed consent, and autonomy [12, 13]. It is a strategy that will be put in place to encourage positive interpersonal relationships between women and health care providers and workers throughout labor, delivery, and the postpartum period [3]. This notion advocates for good staff attitudes, behaviors, and accountability that contribute to women's contentment with their birth experience in a sustainable way [3].

Currently, the change from home delivery to hospital birth has increased access to lifesaving care for difficulties, but it has also generated new challenges, such as facility overcrowding, an excess of procedures, mistreatment, and over-medicalization [14]. Timely, respectful, and

consensual obstetric care, is not the norm in many healthcare settings around the world, especially in developing countries like Ethiopia [14].

Despite a recent dramatic increase in the number of skilled providers and health facilities in Ethiopia, the uptake of prenatal care, skilled delivery service, and postnatal care remain at only 68%, 28%, and 17%, respectively [15]. Even though numerous circumstances contribute to low healthcare utilization, it is becoming evident that poor service quality and provider mistreatment are among the reasons why many women are unable to seek maternal, neonatal, and child health (MNCH) services [16]. Several studies have revealed that women's expectations of how they would be treated at health facilities may have a substantial impact on where they prefer to get maternal health services, notably childbirth [17–19].

In 2016, the Ethiopian government launched its Health Sector Transformation Plan (HSTP), which aims to promote Compassionate and Respectful Care (CRC), with an emphasis on RMC, to improve maternal and newborn health outcomes [20]. Although the target has not yet been met, this plan emphasizes the need of achieving 90 percent skilled birth attendance and lowering the maternal mortality ratio (MMR) from 420/100,000 live births in 2015 to 199/100,000 live births by 2020 [20]. As per small-scale studies conducted in Ethiopia, the prevalence of RMC lies between 12.75% [21] to 77% [22]. Factors affecting the receipt of RMC during childbirth were the place of delivery, time of delivery, ANC uptake, planning status of index pregnancy, educational level, and facing obstetric complications [21–36].

Improving the quality of care through enhancing RMC has been highlighted as the most important intervention for lowering maternal and newborn mortality by laying the path for skilled delivery [5, 6]. Understanding the prevalence and determinants of RMC can help to improve the effectiveness of RMC initiatives and may have a beneficial impact on the uptake of MNCH services [37]. Although several epidemiological studies on the magnitude and determinants of RMC in Ethiopia have been conducted, the results have been inconsistent and varied. Existing studies have also been small-scale or limited by locality, which might also make drawing equivocal conclusions and evidence at the national Prevalence harder. Therefore, such disparities may be inadequate for policymakers and planners to intervene, demanding an assessment of the pooled estimates. Combining information from multiple data sources can enhance estimates of health-related measures by using one source to supply information that is lacking in another. Hence, this systematic review and meta-analysis aimed at estimating the pooled prevalence of RMC and its determinants at the national level. The study's findings will help policymakers and program planners build appropriate interventions to enhance the prevalence of RMC, which is one of the four pillars of HSTP [20].

## Methods

### Study design

While conducting this systematic review and meta-analysis, the Preferred Reporting Items for Systematic Reviews and Meta-Analyses (PRISMA) standards for literature search method, study selection, data extraction, and result reporting were followed [38] (S1 File). To establish the inclusion and exclusion criteria, the PEO (Population, Exposure of interest, Outcome) technique was used, which was adapted from the JBI 2017 review guideline [39].

### Eligibility criteria

**Inclusion criteria.** Population: Women who experienced a childbirth

Exposure of interest: Maternity care (prenatal, skilled delivery, and postnatal) cares

Outcome: Receiving respectful maternity care (RMC).

Study designs: All crossectional studies reporting the prevalence of RMC and its determinants were considered.

Study setting: Community- and facility-based studies conducted in Ethiopia were considered.

Publication status: Both published and unpublished studies were considered, and if a study appeared in multiple reports, the most comprehensive and up-to-date one had been used.

Language: Articles published in the English language were considered.

Year of publication: All publications reported before June 30, 2022, were taken into account.

**Exclusion criteria.**

- Systematic reviews, case series, commentaries, conference abstracts, letters to editors, technical reports, qualitative studies, and other opinion publications were excluded.

- Studies that were not fully accessible after two emails with the primary/corresponding author were excluded since assessing methodological quality in the absence of the full text was problematic.

- Studies that were not explicitly addressed to RMC, such as those studies conducted on CRC in general outpatient department patients, were not taken into account.

- As potential duplicates, studies conducted in the same area during the same study period were excluded.

## Search strategies

The studies had to have been published in English before June 30, 2022. Initially, a comprehensive search of studies was done by using PubMed/MEDLINE, Google Scholar, Science Direct, Scopus, ProQuest, Web of Science, Cochrane Library, and Direct of Open Access Journals. The following keywords were used for the database search: "Respectful", "Woman-Centered", "Dignified", "Friendly", "Non-Abusive", "Compassionate", "Non-discriminatory", "Maternity", "Maternal", "Prenatal", "Antenatal", "Delivery", "Childbirth", "Postnatal", "Care", "Maternal Health Care", "Health Service", "Level", "Magnitude", "Prevalence", "Determinants", "Associated Factors", "Predictors", "Ethiopia", and "Ethiopian". To connect those keywords, Boolean operators (AND and OR) and truncation were employed. The following key search terms were used ("Respectful"[All Fields] OR "Woman-Centered"[All Fields] OR "Dignified"[All Fields] OR "Friendly"[All Fields] OR "Non-Abusive"[All Fields] OR "Compassionate"[All Fields] OR "Non-discriminatory"[All Fields]) AND ("Maternity care"[All Fields] OR "Maternal care"[All Fields] OR "Prenatal care"[All Fields] OR "Antenatal care"[All Fields] OR "Delivery service"[All Fields] OR "Childbirth"[All Fields] OR "Postnatal care"[All Fields] OR "postpartum Care"[All Fields] OR "Maternal Health Care"[All Fields] OR "Health Service"[All Fields] OR "Maternity care"[All Fields]) AND "Ethiopia"[All Fields] (S2 File). Gray literature searches via Google scholar, Google searching, and Addis Ababa and Jimma University institutional repositories supplemented the electronic database search.

## Study selection process

All identified studies were imported into the EndNote XI library and checked for duplication. After removing duplicate articles, three authors (AH, DW, and AT) extracted all articles independently at the title, abstract, and full text. A fourth author (BB) independently reviewed 20%

of the removed studies and compiled the screened articles, and any inconsistencies were settled by discussion. Finally, 16 studies were considered for systematic review and meta-analysis [21–36].

## Data extraction

The data were extracted using a Microsoft Excel spreadsheet. Two authors (AH and DW) separately extracted the important data using a pre-setted and piloted data extraction form. The data extraction format comprised the primary author's name, publication year, study year, study design, study area, study setup, sample size, response rate, data collection technique, the proportion of RMC, and adjusted odds ratio(AOR) with their 95% confidence interval.

## Risk of bias in individual studies

The methodological quality of included studies was assessed by using Joanna Briggs Institute (JBI) Critical appraisal checklist for prevalence studies [40]. Two reviewers independently rated the quality of the included studies (AH and BB). There are nine parameters in the evaluation tool and each parameter has equal weight. (1) Was the sampling frame appropriate to address the target population? (2) Were study participants sampled appropriately? (3) Was the sample size adequate? (4) Were the study subjects and the setting described in detail? (5) Was the data analysis conducted with sufficient coverage of the identified sample? (6) Were valid methods used for the identification of the condition? (7) Was the condition measured in a standard, reliable way for all participants? (8) Was there appropriate statistical analysis? (9) Was the response rate adequate, and if not, was the low response rate managed appropriately? Each item was assessed as either low or high risk of bias. The evaluators assigned a score of '0' if the study met each specific parameter and a score of '1' if it did not. A composite quality index was computed and the risk of bias was graded as low (0–2), moderate (3 or 4), or high (≥5) (S3 File). Articles with low and moderate risks of bias were considered for this systematic review and meta-analysis.

## Measurement of the outcome of interest

The primary outcome variable of this systematic review and meta-analysis was the prevalence of RMC in Ethiopia, which was determined using the pooled prevalence. The secondary outcome variable was RMC determinants, which were estimated using a pooled AOR with 95 percent CIs. RMC is a universal human right that must be provided to all childbearing women in every health system and is measured by four performance standards (friendly care, timely care, discrimination-free care, and abuse-free care). When those women received all four performance domains, they were considered to have received RMC [3, 10, 13, 41].

## Statistical methods and analysis

The data extracted from a Microsoft Excel spreadsheet were exported to the STATA™ 16 statistical software, where all statistical data analyses were undertaken. First, Higgins I-square ($I^2$) statistics and the Cochran's-Q test were used to determine the presence of statistical heterogeneity across included studies. Heterogeneity was classified as low, moderate, or high when the values of I-square were <25, 50–75, and >75%, respectively [40]. Accordingly, significant heterogeneity was detected [$I^2$ = 98.5%, p-value<0.001]. Thus, a random-effects meta-analysis model with the DerSimonian-Laird method was used to determine the pooled prevalence of RMC. The adjusted Odds Ratios(AOR) from eligible studies were extracted, along with their 95% CIs. The pooled AORs were computed using a random- or fixed-effect model. Finally,

forest plots were used to display the pooled estimates for RMC and its determinants, along with their respective 95% confidence intervals.

## Publication bias

The presence of publication bias was visually checked by using funnel plots, and a symmetrical, large inverted funnel revealed that the likelihood of publication bias was less likely. Statistical methods such as Egger's and Begg's tests were used to supplement visual assessment, and a p-value <0.05 indicate the likelihood of publication bias.

## Additional analyses

**Subgroup analyses and heterogeneity.** Subgroup analyses were performed based on geographical regions, residence, study year, and publication year. To identify potential sources of heterogeneity across studies, a univariate meta-regression analysis was performed with sample size, publication years, and study years as covariates.

**Sensitivity analysis.** To assess the influence of a single study on the overall pooled prevalence of RMC, sensitivity analysis was performed using a random-effects model.

# Results

## Study selection

A total of 1599 studies were found through all searches and 1078 records were duplicates and were thus removed. The remaining 521 studies were eligible for screening. Based on the title and abstract screening, 478 studies were excluded, having left 43 full articles. Again, 27 studies were removed (twelve owing to insufficient data, seven failed to state the outcome of interest clearly, two case reports, and six were qualitative studies). Finally, 16 studies were considered for this systematic review and meta-analysis [21–36] (Fig 1).

## Characteristics of included studies

Sixteen studies with a total of 6354 study participants were considered [21–36]. All of the eligible studies were cross-sectional in design. The studies were carried out between 2013 and 2021. All of the included studies collected data through face-to-face interviews with a pre-tested, interviewer-administered questionnaire. Studies conducted in Addis Ababa (n = 173) [36] and South Nations, Nationalities, and Peoples' Region(SNNPR) (n = 783) [29], accounted for the minimum and maximum sample sizes, respectively. In terms of the distribution of the studies across the geographical region, five studies were from Amhara [24, 32–35], five from Oromia [22, 23, 25, 27, 31], two from Addis Ababa [26, 36], two from SNNPR [28, 29], one from Harari [30], and one from Benishangul Gumuz [21]. When it came to the risk of bias in the included studies, the majority (14) had a low risk, with the remaining two having a moderate risk (Table 1).

## The pooled prevalence of RMC in Ethiopia

Because the prevalence estimate varied across studies with significant heterogeneity ($I^2 =$ 98.50%; P<0.001), we used a random-effect model with a DerSimonian and Laird method. The overall pooled prevalence of Respectful maternity care in Ethiopia was found to be 48.44% (95% CI: 39.02–57.87) (Fig 2).

Regarding each component of RMC, 78.53 (95% CI: 72.57, 84.48) and 68.95 (95% CI: 64.52, 73.38) of women received discrimination-free and timely care, respectively. Only half, 49.99 (95% CI: 32.46, 67.52) of women got friendly care(Table 2).

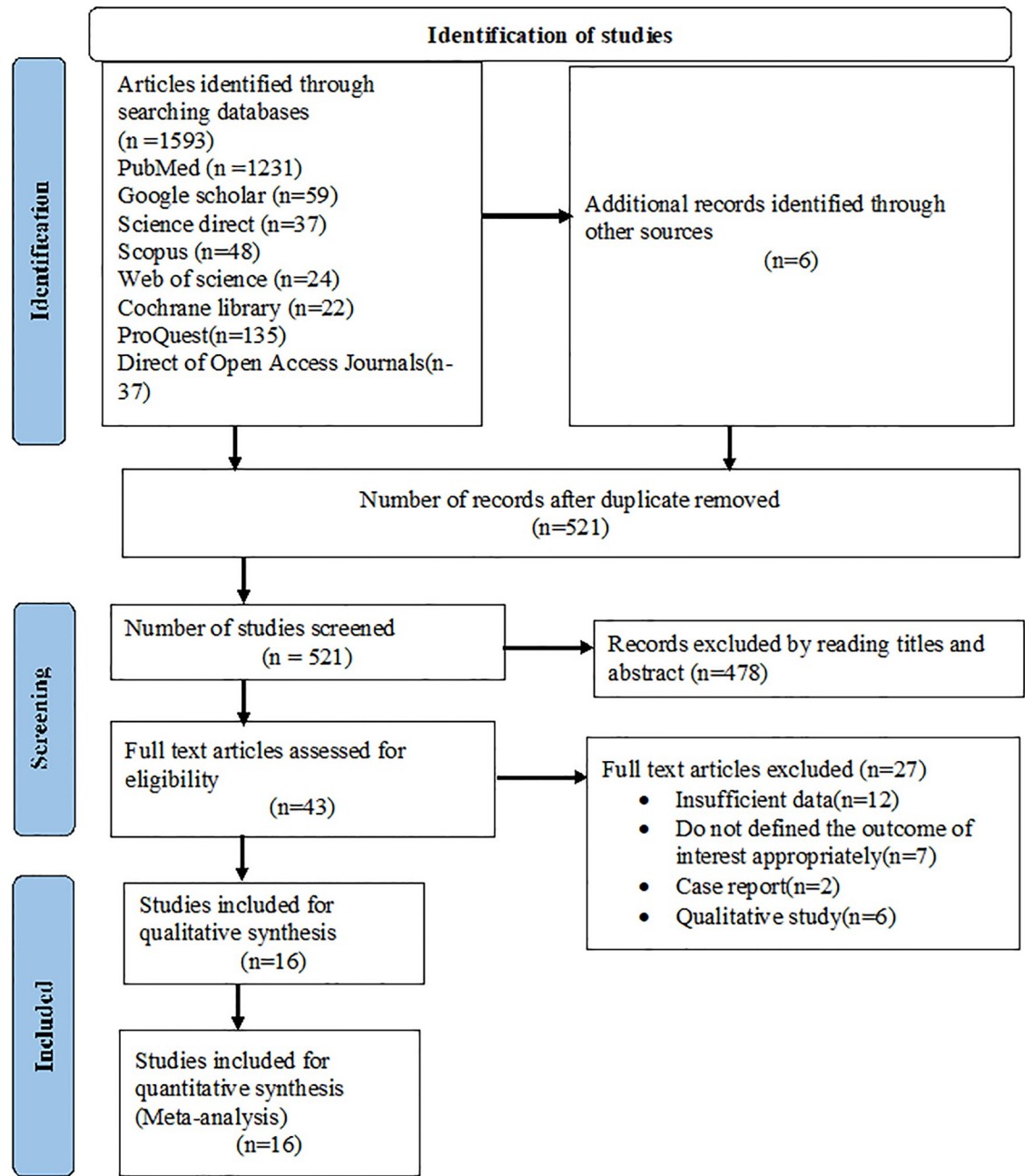

**Fig 1. PRISMA flow diagram describing the selection of studies for systematic review and meta-analysis.**

## Subgroup analyses

Subgroup analyses were conducted by region, study year, and publication year. Accordingly, the highest prevalence of RMC was reported in Oromia and Amhara regions, 58.01%(95% CI: 42.44, 73.58), and 51.31%(95% CI: 41.10, 61.53), respectively. On the other hand, the lowest Prevalence of RMC was reported in Benishangul Gumuz, 12.65% (95% CI: 9.45, 15.90) (Fig 3).

In addition, we performed a subgroup analysis based on the year when the studies were conducted. Accordingly, the pooled prevalence of RMC was 41.15% (95% CI: 28.85–53.44) for

**Table 1. Descriptive summary of studies included in systematic review and meta-analysis of the prevalence of RMC and its determinants in Ethiopia, 2015–2022.**

| Authors name, year of publication | Study year | Region | Study area | Study design | sampling techniques | Sample size | Response rate | RMC | Risk of bias |
|---|---|---|---|---|---|---|---|---|---|
| Amsalu et al., 2022 [21] | 2019 | Benishangul Gumuz | Benishangul Gumuz | CS | srs | 404 | 97.34 | 12.65 | Low |
| Yismaw et al., 2022 [23] | 2019 | Oromia | Illu ababora | CS | SRS | 281 | 98.5 | 47.30 | Low |
| Yalew et al., 2022 [24] | 2018 | Amhara | Dessei | CS | srs | 389 | 99.7 | 43.40 | Low |
| Eneyew et al., 2021 [22] | 2021 | Oromia | Jimma | CS | srs | 348 | 100 | 77.00 | Moderate |
| Adane et al., 2021 [25] | 2019 | Oromia | Shashemene | CS | srs | 420 | 99.5 | 63.00 | Low |
| Ambachew, 2021 [26] | 2021 | Addis Ababa | A.A | CS | srs | 384 | 99.2 | 65.81 | Low |
| Cafo et al., 2021 [27] | 2020 | Oromia | Wollega | CS | SRS | 351 | 91.4 | 66.95 | Low |
| Abdo et al., 2021 [28] | 2020 | SNNPR | Hadiya | CS | srs | 413 | 97.86 | 53.00 | Moderate |
| Wochefu et al, 2021 [29] | 2019 | SNNPR | Hawassa | CS | srs | 783 | 97.11 | 36.50 | Low |
| Bante et al., 2020 [30] | 2017 | Harari | Harar | CS | srs | 425 | 100 | 38.40 | Low |
| Bulto et al., 2020 [31] | 2018 | Oromia | West Shoa Zone | CS | srs | 567 | 97.5 | 35.80 | Low |
| Yosef et al., 2020 [32] | 2020 | Amhara | Northwest Amhara | CS | srs | 410 | 97.16 | 56.30 | Low |
| Wubetu et al., 2020 [33] | 2019 | Amhara | Debre Birhan | CS | SRS | 412 | 99.8 | 35.70 | Low |
| Dagnaw et al., 2020 [34] | 2019 | Amhara | Dessei town | CS | SRS | 310 | 97.8 | 64.50 | Low |
| Wassihun and Zeleke, 2018 [35] | 2017 | Amhara | Bahirdar | CS | srs | 284 | 100 | 57.0 | Low |
| Asefa and Bekele, 2015 [36] | 2013 | Addis Ababa | Addis Ababa | CS | SRS | 173 | 100 | 22.00 | Low |
| Total | | | | | | 6354 | 98.12 | | |

CS: Cross-sectional study, SRS: systematic random sampling, srs: simple random sampling

studies conducted before 2020 and 68.83% (95% CI: 55.30–72.37) for studies conducted in 2020 and after (Fig 4).

## Heterogeneity and publication bias

A univariate meta-regression analysis was run using study-Prevalence characteristics (publication year and sample size) as a cofactor to identify the possible source of heterogeneity across the included studies. However, heterogeneity was not explained by sample size (P = 0.582), and the publication year (P = 0.448) (Table 3).

The funnel plot was used to visually examine publication bias, and the effect estimates were asymmetrical, indicating that publication bias was unlikely (Fig 5). Furthermore, we checked the presence of publication bias statistically by running Egger's regression test and an adjusted Beggs rank correlation test and the p values were 0.51 and 0.86, respectively. All of these indicate that the presence of publication bias in this study was unlikely.

## Sensitivity analysis

A sensitivity analysis using a random-effects model was carried out to detect the effects of a single study on the overall meta-analysis estimate. As a result, there is no evidence that a single study influenced the pooled prevalence of RMC (Fig 6).

## Determinants of RMC

Thirteen variables were extracted from the included studies to identify determinants of RMC (S4 File). As significant determinants of RMC, six variables were identified namely giving

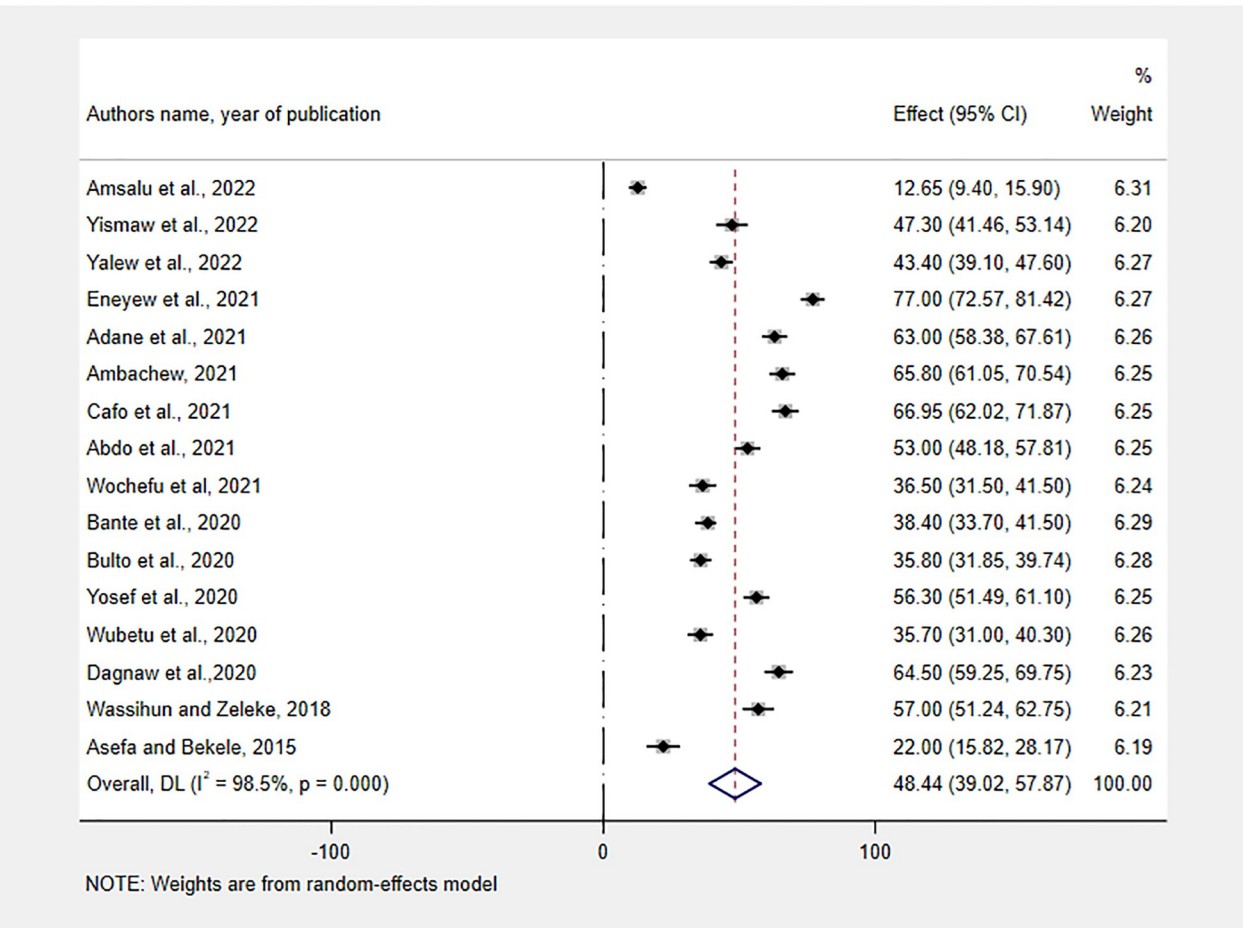

**Fig 2. Forest plot showing the pooled estimates of RMC in Ethiopia, 2013–2022.**

birth during the day, planning status of the last pregnancy, having adequate ANC visit, experiencing an obstetric complication during the last delivery, and receiving service from health care providers who were trained on CRC.

The influence of CRC training on RMC was assessed by using the findings of three studies [21, 23, 31]. Women who received maternal health services from CRC-trained healthcare providers were 4.09 times more likely than their counterparts to receive RMC [AOR = 4.09, 95% CI: 1.73, 6.44] (Fig 7).

Moreover, we used four studies [30–33] to assess the relationship between having ANC and receiving RMC during childbirth. Accordingly, women who received adequate ANC have a 2.34 times greater chance of receiving RMC than their counterparts [AOR = 2.34, 95% CI: 1.62, 3.06] (Fig 8).

The effect of time when childbirth took place was assessed using the findings of five studies [21, 24, 31–33]. A fixed-effect meta-analysis of AORs revealed that the odds of receiving RMC were 2.61 times higher for women who gave birth during the daytime as compared to those who gave birth at night [AOR: 2.61, 95% CI: 1.92, 3.31] (Fig 9).

As per the findings of four studies [23, 25, 30, 31], the pregnancy planning status at the time of childbirth had a positive association with RMC. The likelihood of receiving RMC was 4.43

**Table 2. The pooled prevalence of domains of RMC in Ethiopia, 2022.**

| Domains of RMC with a list of studies | Sample size | Pooled prevalence(95% CI) | Heterogeneity | | P-value |
|---|---|---|---|---|---|
| | | | $I^2$ | Cochran's Q | |
| **Friendly care** | 3175 | 49.99 (32.46,67.52) | 99.2 | 899.01 | 0<0.001 |
| Amsalu et al., 2022 [21] | | | | | |
| Yismaw et al., 2022 [23] | | | | | |
| Adane et al., 2021 [25] | | | | | |
| Ambachew, 2021 [26] | | | | | |
| Bante et al., 2020 [30] | | | | | |
| Bulto et al., 2020 [31] | | | | | |
| Yosef et al., 2020 [32] | | | | | |
| Wassihun and Zeleke, 2018 [35] | | | | | |
| **Abuse-free care** | 3958 | 58.36(46.44,70.29) | 98.5 | 537.34 | 0<0.001 |
| Amsalu et al., 2022 [21] | | | | | |
| Yismaw et al., 2022 [23] | | | | | |
| Adane et al., 2021 [25] | | | | | |
| Ambachew, 2021 [26] | | | | | |
| Wochefu et al, 2021 [29] | | | | | |
| Bante et al., 2020 [30] | | | | | |
| Bulto et al., 2020 [31] | | | | | |
| Yosef et al., 2020 [32] | | | | | |
| Wassihun and Zeleke, 2018 [35] | | | | | |
| **Timely care** | 3958 | 68.95(64.52, 73.38) | 89.4 | 75.82 | 0<0.001 |
| Amsalu et al., 2022 [21] | | | | | |
| Yismaw et al., 2022 [23] | | | | | |
| Adane et al., 2021 [25] | | | | | |
| Ambachew, 2021 [26] | | | | | |
| Wochefu et al, 2021 [29] | | | | | |
| Bante et al., 2020 [30] | | | | | |
| Bulto et al., 2020 [31] | | | | | |
| Yosef et al., 2020 [32] | | | | | |
| Wassihun and Zeleke, 2018 [35] | | | | | |
| **Discrimination- free care** | 3958 | 78.53(72.57, 84.48) | 96.1 | 207.03 | 0<0.001 |
| Amsalu et al., 2022 [21] | | | | | |
| Yismaw et al., 2022 [23] | | | | | |
| Adane et al., 2021 [25] | | | | | |
| Ambachew, 2021 [26] | | | | | |
| Wochefu et al, 2021 [29] | | | | | |
| Bante et al., 2020 [30] | | | | | |
| Bulto et al., 2020 [31] | | | | | |
| Yosef et al., 2020 [32] | | | | | |
| Wassihun and Zeleke, 2018 [35] | | | | | |

times higher for those mothers with planned pregnancies as compared to women with unplanned pregnancies [AOR: 4.43, 95% CI: 2.74, 6.12] (Fig 10).

Finally, a negative association was identified between having obstetric complications and RMC. Those women who sustained any obstetric complication had 54% less likely to get RMC than those women who didn't face any obstetric complication [AOR: 0.46, 95% CI: 0.30, 0.61] (Fig 11).

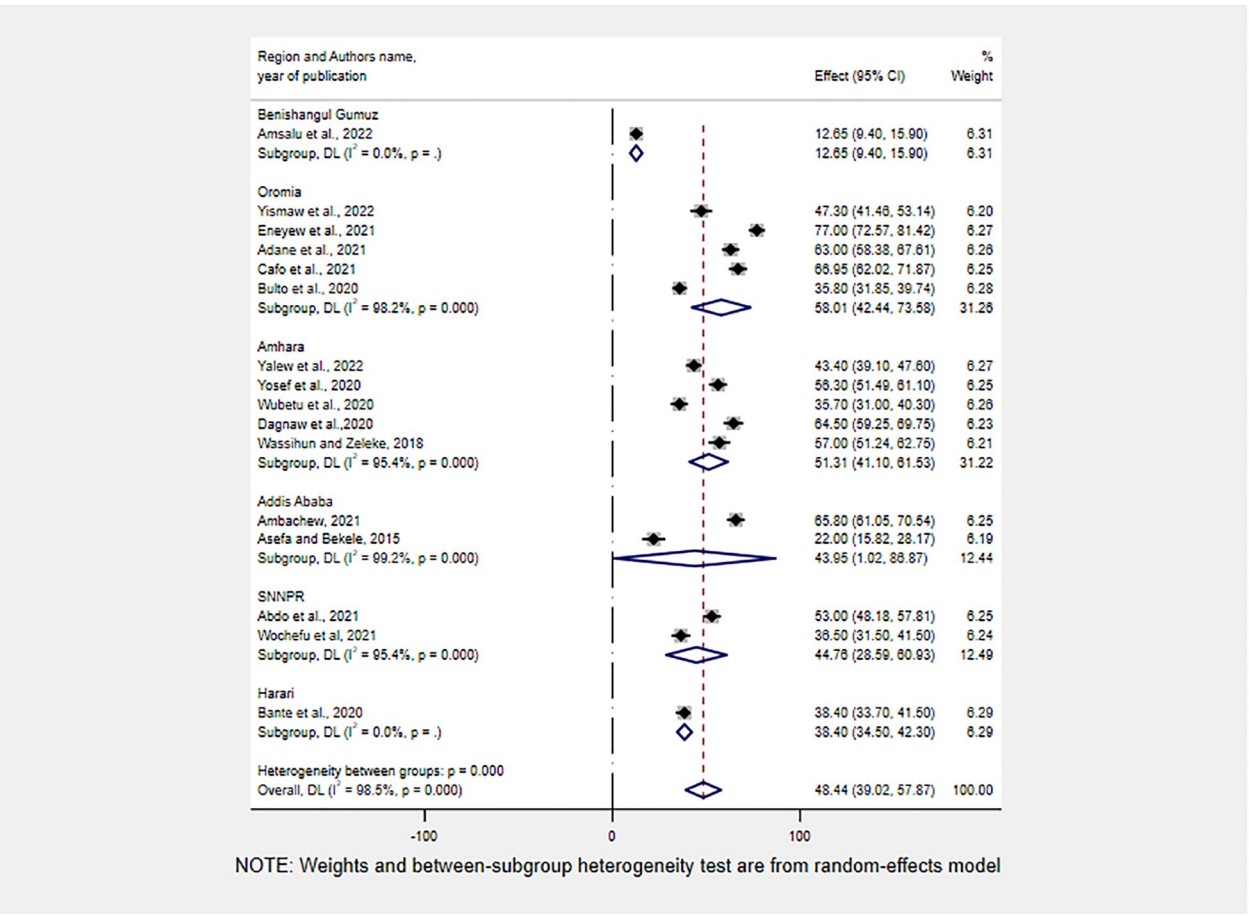

**Fig 3. Sub-group analysis for the pooled prevalence of RMC by geographical regions of Ethiopia, 2013–2022.**

## Discussion

Ethiopian HSTP-I and HSTP-II advocate ensuring equitable and timely delivery of quality health care (reliable, patient-centered, and efficient) to all in need [20, 42]. Safe motherhood must include respect for women's basic human rights, such as autonomy, decency, feelings, preferences, and priorities, in addition to the prevention of illness or death [5, 10]. Currently, WHO recommends providing RMC per the human rights-based approach to reducing maternal and newborn morbidity and mortality by improving women's pregnancy and childbirth experiences and addressing inequities in MNCH care access [6]. Given the importance of RMC in ensuring the quality of MNCH services, assessing its status at the national Prevalence allows for a better understanding of its potential contribution to the national and global accomplishment of HSTP [42] and SDG [8, 9], respectively. Hence, this systematic review and meta-analysis was aimed at determining the Prevalence of RMC and its determinants in Ethiopia.

The estimated pooled prevalence of RMC in Ethiopia was 48.44% (95% CI: 39.02–57.87). Accordingly, the Prevalence of RMC was higher as compared to findings of a systematic review and meta-analysis in India(28.7%) [43] and a study conducted in East and Southern Africa (30%) [44]. In addition, the current finding was higher than primary studies conducted in Pakistan(2.6% and 0.5%) [45, 46], Peru(2.6%) [47], Tanzania(30%) [48], and Nigeria(2%) [49].

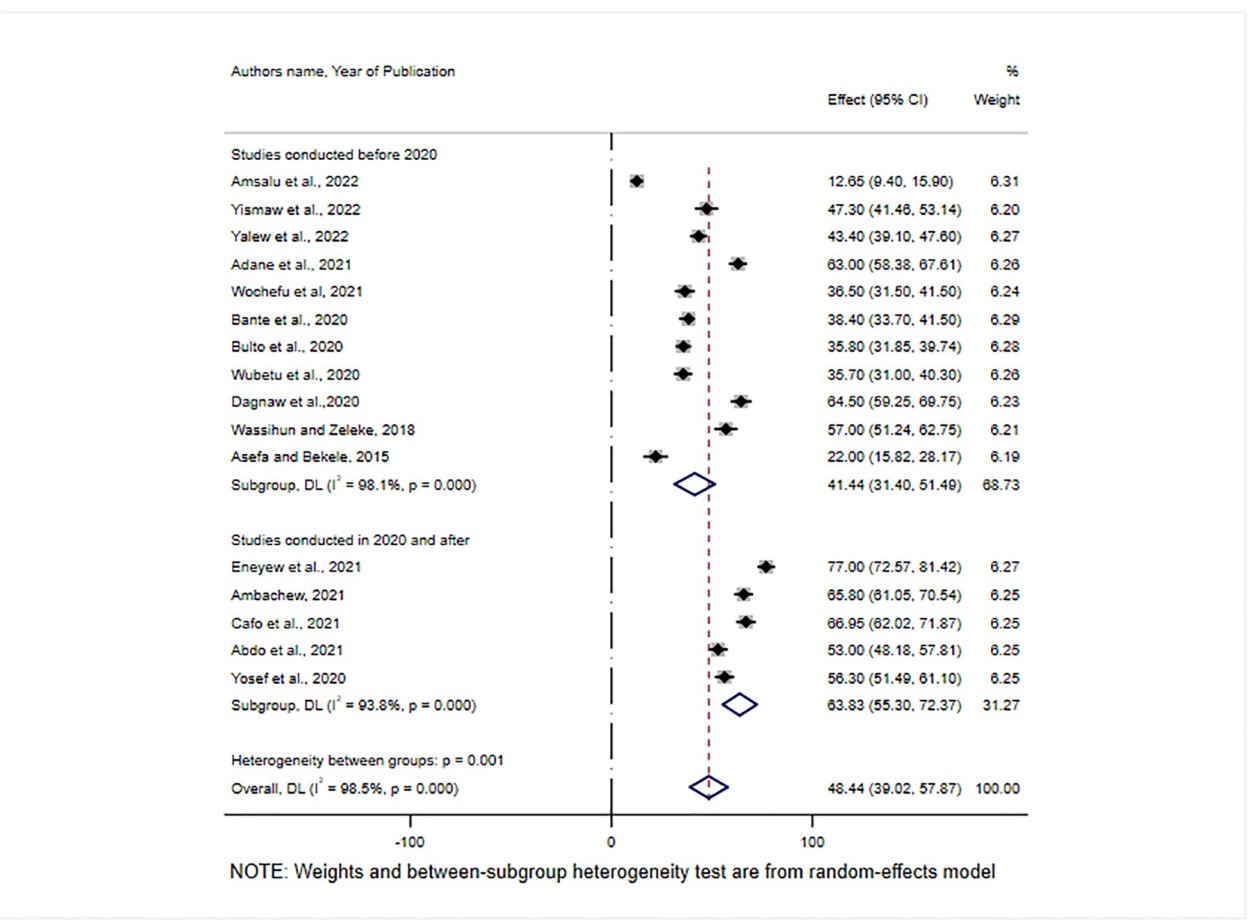

**Fig 4. Sub-group analysis for the pooled prevalence of RMC by study year in Ethiopia, 2022.**

This could be due to the Ethiopian government's effort since 2020 to create CRC healthcare providers as one of the pillars of HSTP-I and–II [20, 42]. Furthermore, there is an increasing commitment and interest in implementing the CRC initiative at the national prevalence through the provision of CRC training to over 27, 000 health leaders and health workers across the country, with a particular emphasis on those who work at MNCH service delivery points [50].

On the other hand, the finding was lower than primary studies conducted in Brazil (81.7%) [51], Mexico (72.3%) [30], India(76%) [52], and Kenya(80.0%) [53]. The disparity in these findings could be attributed to differences in the methodologies and tools used to measure RMC, socio-cultural and economic differences, study period, and organizational factors such as a shortage of health facility to population ratio. Furthermore, it was significantly lower than the Ethiopian government's stated goal of increasing CRC to 90 percent by 2025 in the

**Table 3. A univariate meta-regression analysis of factors affecting between-study heterogeneity.**

| Heterogeneity source | Coefficients | Std.Err | p-value | 95% CI |
|---|---|---|---|---|
| Sample size | -0.0198533 | 0.036104 | 0.582 | -.0906158, .0509091 |
| Publication year | 2.237802 | 2.948367 | 0.448 | -3.540891, 8.016496 |

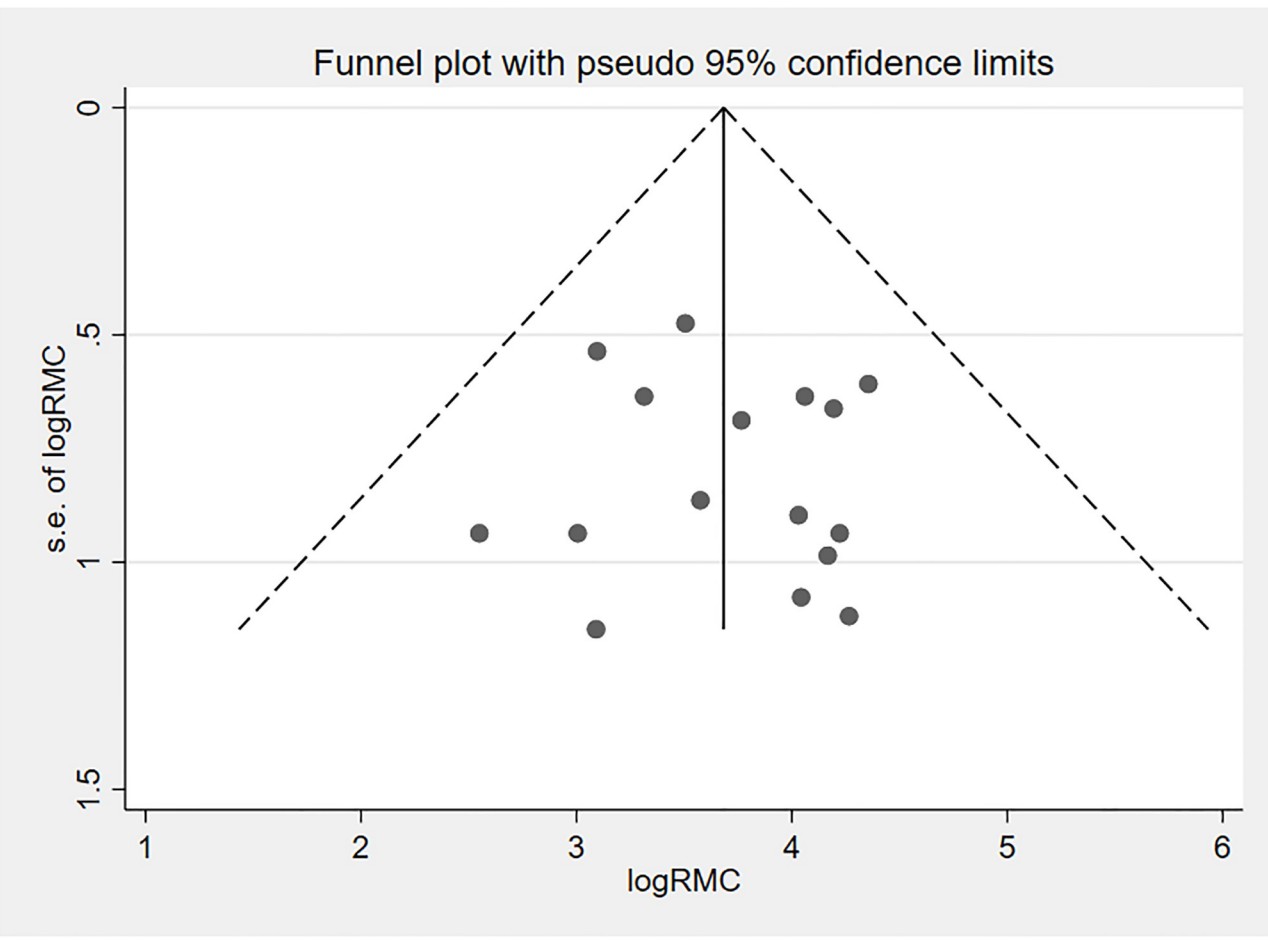

**Fig 5. Funnel plot displaying publication bias of studies reporting the RMC in Ethiopia, 2022.**

HSTP-II [42]. Also, the current finding was found to be low as compared to a finding obtained by direct observation of 16 model health facilities in Ethiopia(60.4%) [54]. Hence, the government should work to ensure an adequate number and mix of quality health workforces who are motivated, competent, and compassionate to enhance RMC. In addition, due consideration should be given to the development of a short-term training manual that improves the awareness and practice of RMC among health workers at service delivery points [42]. The Ministry of Health should implement a multi-pronged strategy, starting with the enrollment of students in health science programs and the efficient administration of currently employed health professionals [55].

As per subgroup analysis results, the highest and the lowest Prevalence of RMC was reported in the Oromia, 58.01%(95% CI: 42.44, 73.58) and Benishangul Gumuz region 12.65% (95% CI: 9.45, 15.90) respectively. These differences could be attributed to regional disparities in the number of HCPs and health facilities, with the region with the lowest prevalence being one of Ethiopia's emerging regions with the lowest MNCH service coverage [15]. Furthermore, the low prevalence could be attributed to the small number of studies included in this meta-analysis, which included only one study from the Benishangul region. Furthermore, in the last 2–3 years, the Benishangul Gumuz region has been one of the most insecure, with frequent conflicts that have resulted in the displacement of civilians and healthcare providers due to

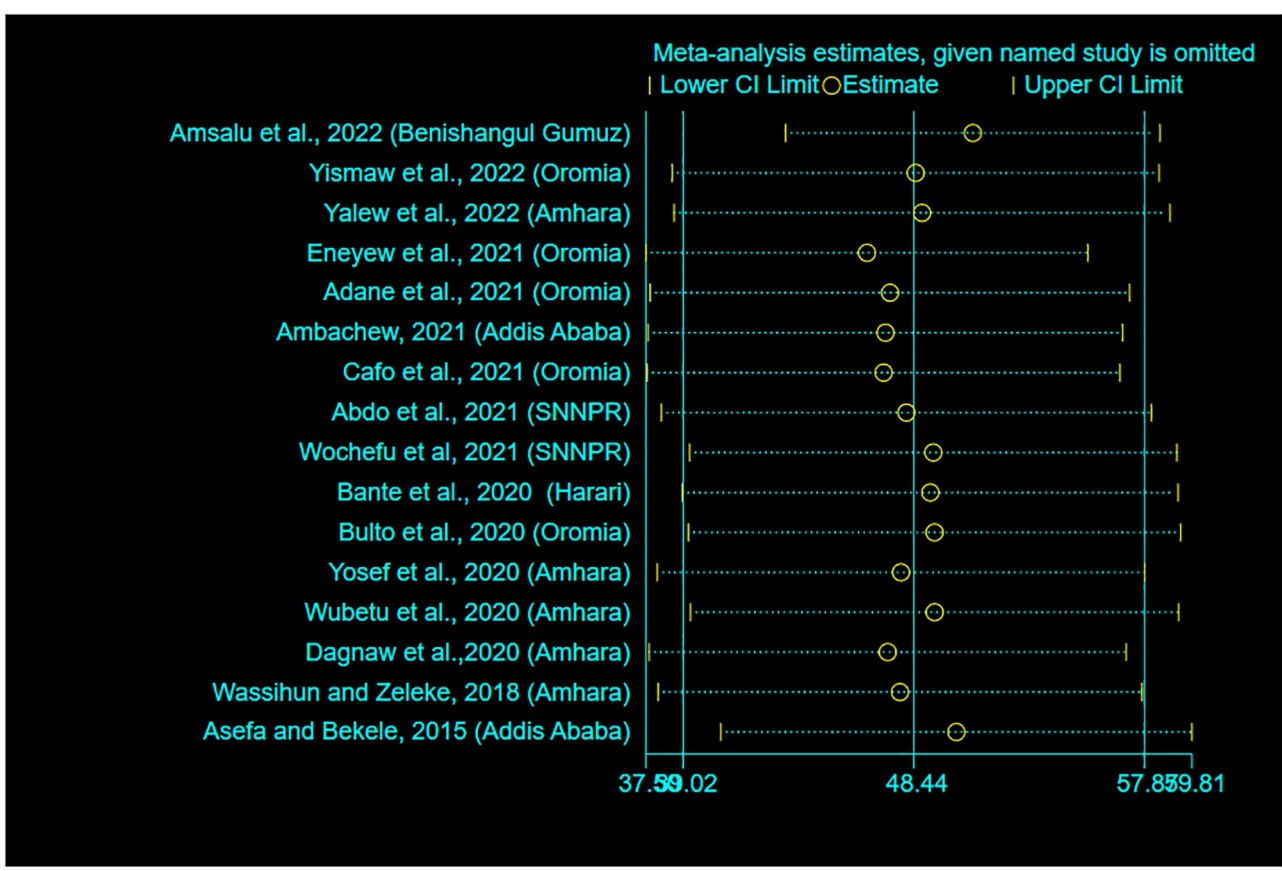

**Fig 6. Sensitivity analysis for the pooled prevalence of RMC in Ethiopia, 2022.**

security concerns [56]. As a result, there may be a lapse in stringently monitoring the MNCH program, resulting in low RMC.

Furthermore, from a sub-group analysis studies conducted since 2020 had the highest Prevalence of RMC, 63.83% (95% CI:55.3, 72.37), compared to studies conducted before 2020. The possible justification could be the Ethiopian government's emphasis on addressing a low uptake of maternal health service utilization by improving RMC through various measures, particularly in the previous two years [42]. The measures taken were, creating model professionals in each health facility, advocacy campaigns through mass media, and enacting a Patients' Rights and Responsibilities law [42, 50]. Furthermore, the development and implementation of a generic curriculum in pre-service education, as well as the establishment of well-functioning 16 CRC incubation centers, including national referral and regional hospitals, may have contributed to the good progress of the RMC Prevalence over the last three years [55].

Another objective of this systematic review and meta-analysis was to identify the most important factors that affect the Prevalence of RMC. Accordingly, receiving service from CRC-trained health care providers, having adequate ANC visits, planning status of the last pregnancy, giving birth during the daytime, and experiencing an obstetric complication were identified as determinants of RMC.

According to the current systematic review and meta-analysis, receiving MNCH services from CRC-trained providers increases the likelihood of receiving RMC. The finding was supported by studies conducted in India [57, 58], Sanford, USA [59], and Tanzania [60]. It is

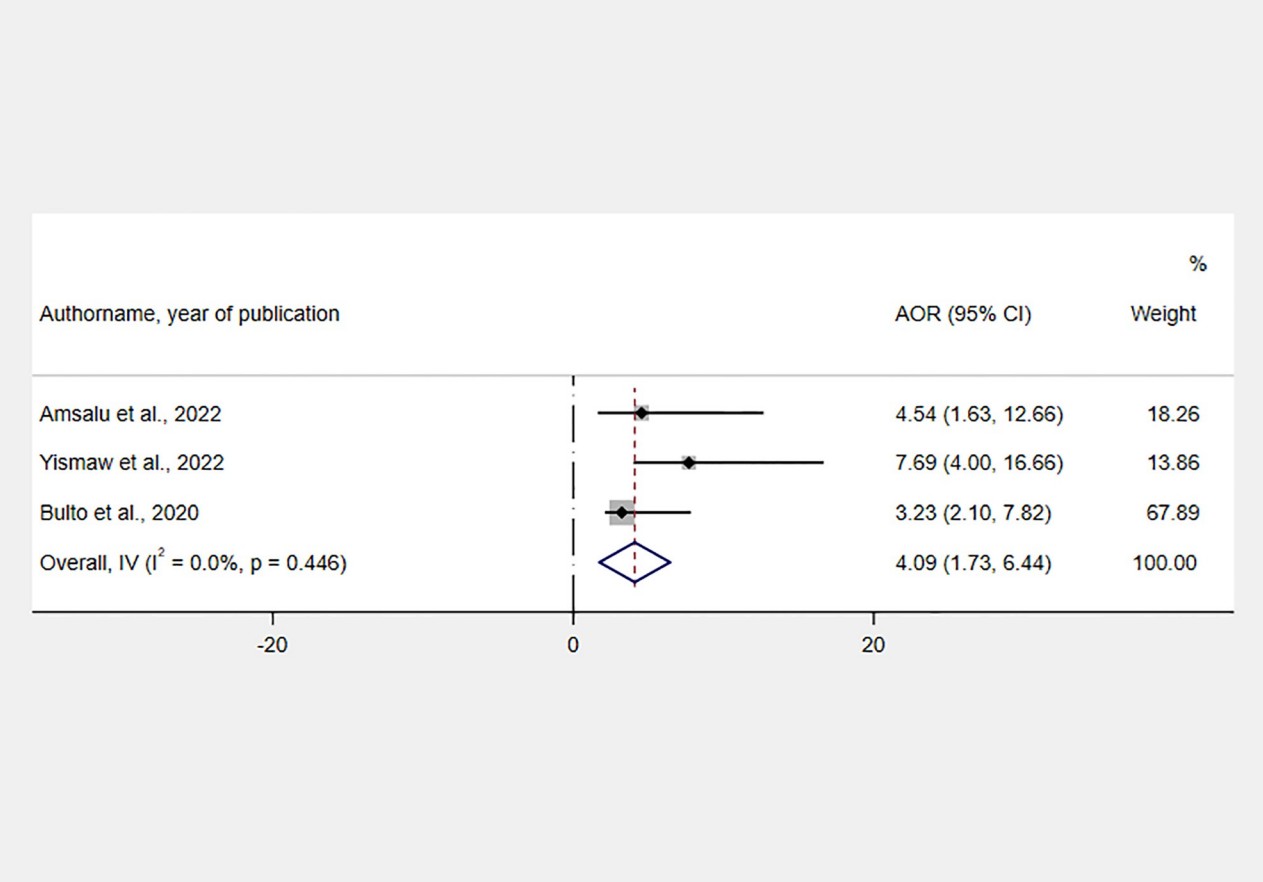

**Fig 7. Forest plot showing the association between service delivered by CRC-trained healthcare providers and RMC in Ethiopia, 2022.**

widely acknowledged that CRC training is vital for gearing up MNHC providers to offer human-centered care, serve patients ethically and with respect, keep taking a professional oath, and promote providers to provide clients with satisfactory service quality [20, 59, 61]. Besides that, the training may influence HCPs' knowledge, motivation, and attitude toward the RMC, which will have a significant positive impact on its provision. As a result, managers in the health sector need to emphasize on the provision of CRC training for health care providers, with a due consideration paid to those who work at maternity service delivery points. The Federal Ministry of Health should collaborate with the Ministry of Education to incorporate CRC issues into the acting curriculum in order to familiarize newly emerging health care providers with the RMC.

In addition, women who received adequate ANC had a greater chance of receiving RMC. This finding was in tandem with studies conducted in Kenya [53] and Tanzania [48]. The possible justification could be that women who had adequate ANC visits had a better chance of acclimating to the health facility setup and developing close relations with the HCP. As the evidence showed that having adequate ANC may result in a change in the dynamic between provider and client, which may increase the likelihood of receiving RMC [62]. All of these are essential in ingraining trust in the facility's services, which resulted in RMC [31, 50].

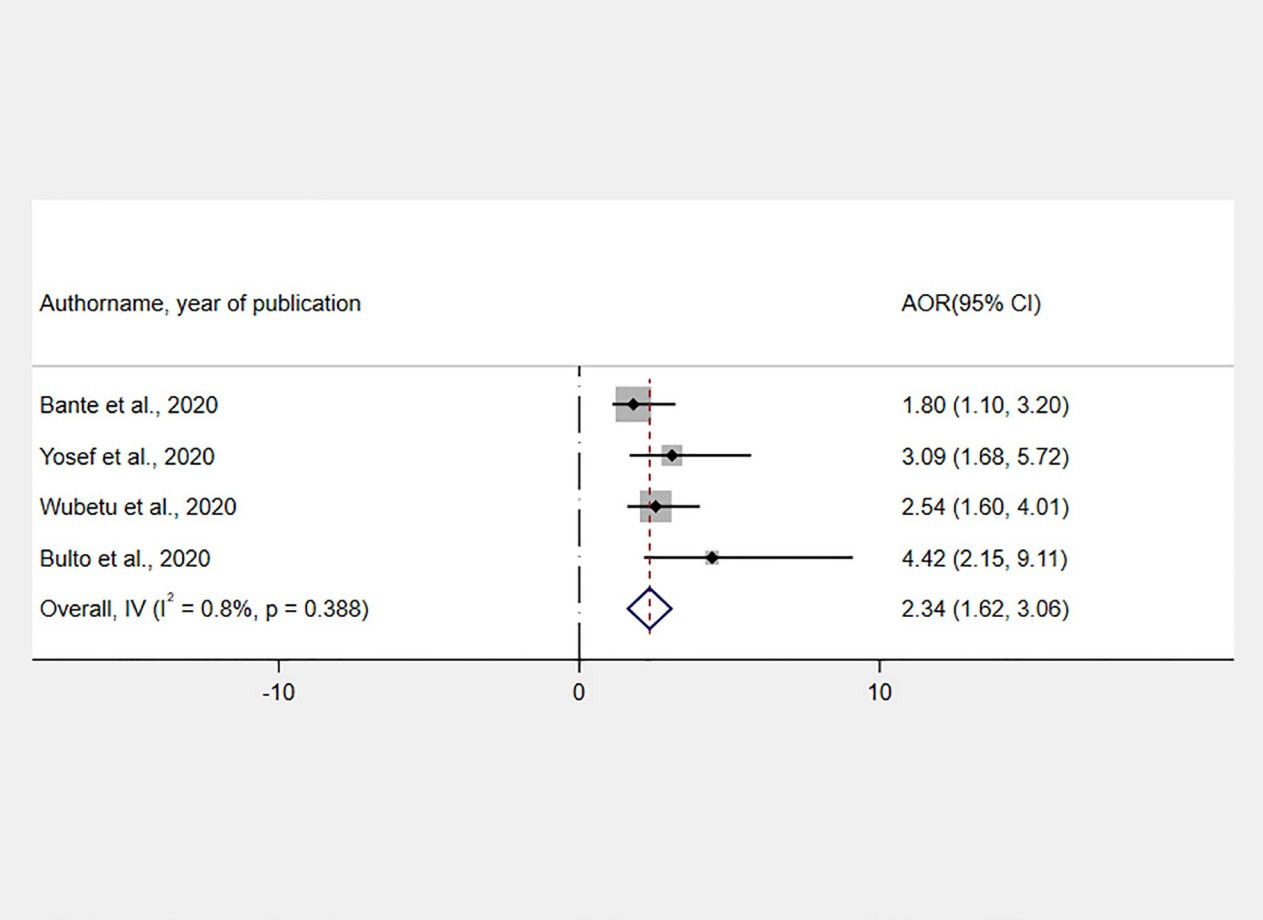

**Fig 8. Forest plot showing the association between ANC and RMC during childbirth in Ethiopia, 2022.**

Furthermore, women with planned pregnancies were more likely to receive RMC than those who had not. This could be because women with planned pregnancies are more likely to receive prenatal care services in the same facility where they will give birth, facilitating their interaction with health professionals and ultimately leading to RMC [48]. Furthermore, using MNCH services helps women become acquainted with the service providers, reduces depression, and increases the mother's attitude toward the care as respectful. Evidence indicated that planned pregnancy increases women's contentment and they acknowledge the service provided as reverent [63]. On the other perspective, HCPs should be aware that it is their responsibility to treat all birthing women equally, regardless of their pregnancy planning status. Rather than mistreating women who have had unplanned pregnancies, it would be advisable to focus on preventing those very pregnancies through the provision of contraception.

In addition, this meta-analysis discovered that women who gave birth during the day had a higher chance of receiving RMC than those who gave birth at night. The finding was supported by studies conducted in Kenya [64]. This could be because there are more healthcare providers during the day than at night, when only one health worker may be assigned to duty in health centers. Furthermore, the way senior health workers and managers monitor health care providers during the day may be good, which gives rise to the delivery of RMC. At the national level, the majority of health facilities had infrastructural problems, such as a lack of electricity

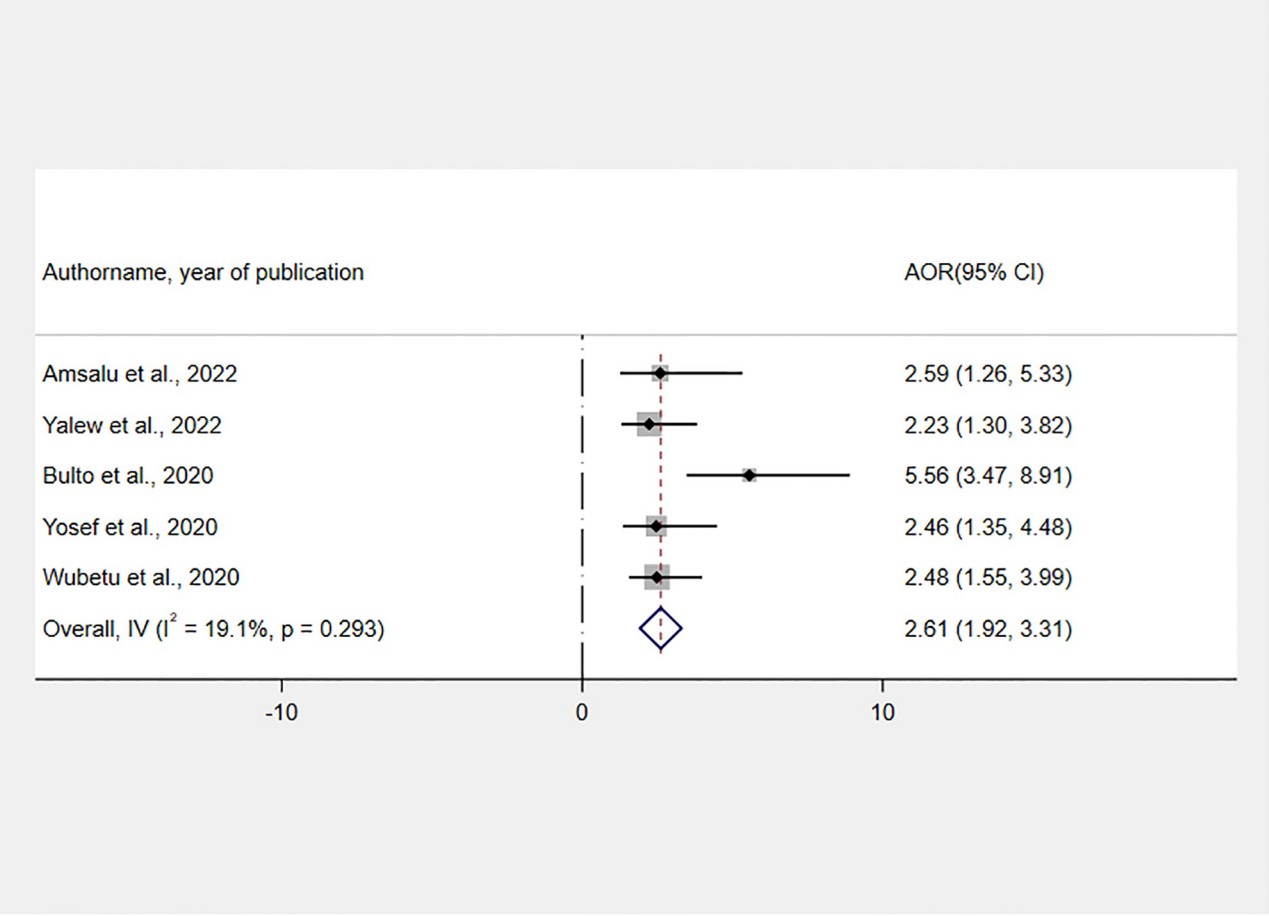

**Fig 9. Forest plot showing the association between daytime delivery and RMC in Ethiopia, 2022.**

[65], and the range of this problem may be lowered during daytime childbirth, enhancing the likelihood of receiving RMC [50]. On the other hand, the tendency to receive low RMC during the night shift may be explained by the low staff number -to obstetric cases that require night-time maternity care services (i.e labor starts for most women at night time) [21, 66]. Furthermore, health providers may become tired during the night due to workload, and they may not act normally because they are awake from sleep, all of which may result in physical or verbal abuse of the parturient [32]. This may be an implication for human resource managers to assign an adequate number of HCPs to the night-shift duties to reduce workload.

Finally, women who experienced obstetric complications were found to have a lower likelihood of receiving RMC. Studies conducted in India [57] and Tanzania [18] corroborated the current finding. This could be because women who experienced complications during labor are more likely to develop postpartum blues and depression, which can impede and lower the process and prevalence of receiving RMC [57]. In addition, complicated labor necessitates frequent and meticulous follow-up, which exhausts the provider and may result in service abandonment. Furthermore, those women are admitted and stay in health facilities for an extended period with little or no support, and they may perceive the service as unwelcoming, which may result in underreporting of RMC. Furthermore, there are several dimensions of D&A that could theoretically be associated with complicated birth (e.g., unconsented care, lack of

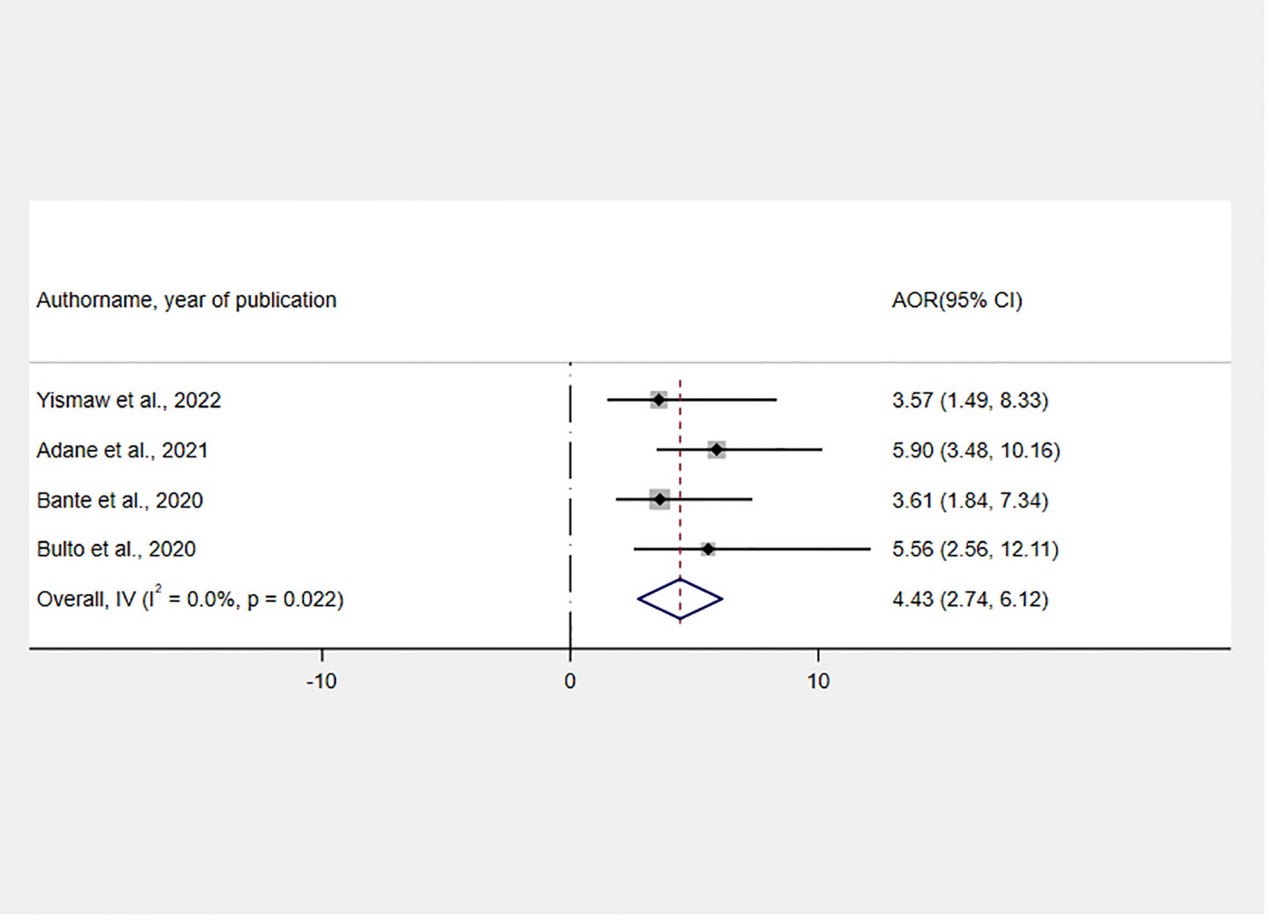

**Fig 10. Forest plot showing the association between planned pregnancy and RMC in Ethiopia, 2022.**

information and choice, lack of respect for values and preferences, exclusion of choice companion, lack of privacy), and all of these could result in a low prevalence of RMC [62].

Regarding the strength, this was the first systematic review and meta-analysis of its kind in Ethiopia to assess the prevalence of RMC and its determinants. It could help policymakers and managers at all levels to improve the quality of MNCH, which is one of the HSTP and SDG agendas [8, 42]. However, due to some of the limitations listed below, the findings should be interpreted with caution. First, the search only included articles published in English. Because of the nature of the study design, the majority of the studies considered were cross-sectional, making it difficult to establish a cause-effect relationship. Furthermore, the studies were limited to six regions, which may limit the generalizability of the findings. Finally, because of the scarcity of comparable systematic reviews and meta-analyses, we were compelled to discuss some of our findings, with primary studies conducted outside of Ethiopia.

## Conclusion

As per this meta-analysis, the Prevalence of RMC in Ethiopia was low, suggesting that more emphasis is needed to plan and implement intervention measures. The pooled prevalence of receiving RMC varied across geographical regions and study periods. Accordingly, receiving service from CRC-trained health care providers, having ANC visits, pregnancy planning

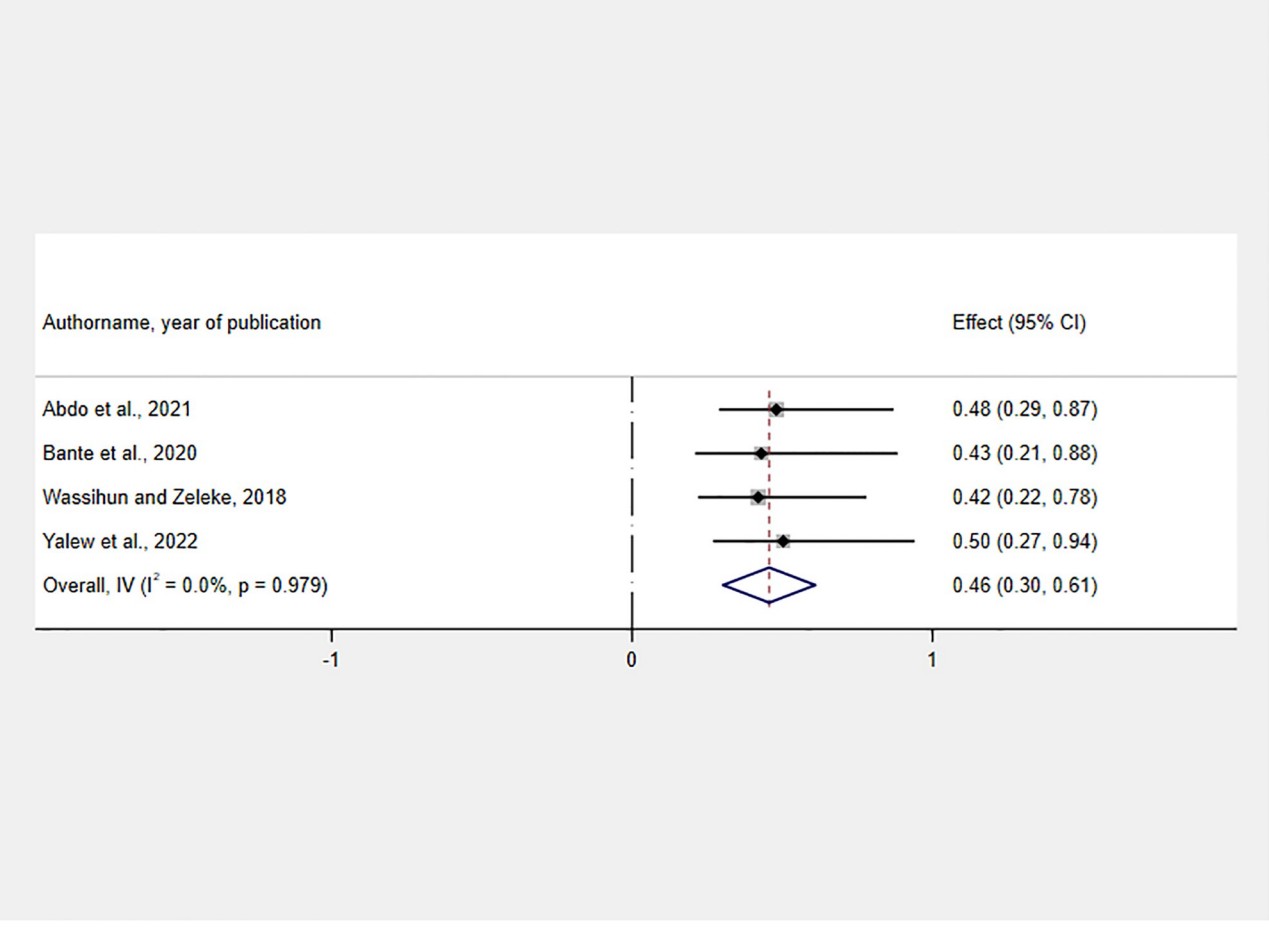

**Fig 11. Forest plot showing the association between facing obstetric complications and RMC in Ethiopia, 2022.**

status, giving birth during the daytime, and experiencing an obstetric complication were identified as determinants of RMC. Managers in the health sector need to give due emphasis to the provision of CRC training for healthcare providers, who work at maternity service delivery points. Stakeholders in the health sector need to work to increase the uptake of prenatal care to improve client-provider relationships across a continuum of care. Human resource managers should assign an adequate number of HCPs to the night-shift duties to reduce the workload among obstetric providers. Due emphasis needs to be given to those women who developed an obstetric complication through continuous follow-up.

## Supporting information

**S1 File. PRISMA checklist 2020 used to report the result of systematic review and meta-analysis.**
(DOCX)

**S2 File. Examples of the search strategy for systematic review and meta-analysis on the Prevalence of RMC and its determinants in Ethiopia, 2022.**
(DOCX)

**S3 File. JBI critical appraisal checklist for prevalence studies used for assessing the individual quality of all studies included in the systematic review and meta-analysis, 2022.**
(DOCX)

**S4 File. List of variables considered for estimation of pooled odds ratio.**
(XLSX)

**S5 File. Minimal data set that is used to estimate the pooled prevalence.**
(DTA)

## Acknowledgments

We would like to thank Wachemo University, College of Medicine and Health Sciences, for providing us with free internet access while we were conducting this research. We would like to express our gratitude to all of the authors of the studies included in this systematic review and meta-analysis.

## Author Contributions

**Conceptualization:** Aklilu Habte.

**Data curation:** Aklilu Habte.

**Formal analysis:** Aklilu Habte, Aiggan Tamene, Demelash Woldeyohannes.

**Investigation:** Aklilu Habte, Fitsum Endale, Biruk Bogale, Addisalem Gizachew.

**Methodology:** Aklilu Habte, Aiggan Tamene, Demelash Woldeyohannes, Addisalem Gizachew.

**Software:** Aklilu Habte.

**Supervision:** Aklilu Habte, Fitsum Endale.

**Validation:** Aklilu Habte.

**Visualization:** Aklilu Habte, Biruk Bogale.

**Writing – original draft:** Aklilu Habte, Aiggan Tamene.

**Writing – review & editing:** Aklilu Habte, Aiggan Tamene, Demelash Woldeyohannes, Fitsum Endale, Biruk Bogale.

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
