## [Decision Letter · Decision Letter 0]

2 Oct 2022

PONE-D-22-19516Towards the quality of maternal and newborn health care: The level and determinants of respectful maternity care during childbirth in Ethiopia: A systematic review and Meta-analysisPLOS ONE

Dear Mr Aklilu Habte,

Thank you for submitting your manuscript to PLOS ONE. After careful consideration, we feel that it has merit but does not fully meet PLOS ONE’s publication criteria as it currently stands. Therefore, we invite you to submit a revised version of the manuscript that addresses the points raised during the review process.

We look forward to receiving your revised manuscript.

Kind regards,

Zemenu Yohannes Kassa, Msc

Academic Editor

PLOS ONE

Journal Requirements:

2. Please upload a new copy of Figures 3 and 4 as the detail is not clear. Please follow the link for more information: " ext-link-type="uri" xlink:type="simple">https://blogs.plos.org/plos/2019/06/looking-good-tips-for-creating-your-plos-figures-graphics/"
https://blogs.plos.org/plos/2019/06/looking-good-tips-for-creating-your-plos-figures-graphics/

3. We note that this manuscript is a systematic review or meta-analysis; our author guidelines therefore require that you use PRISMA guidance to help improve reporting quality of this type of study. Please upload copies of the completed PRISMA checklist as Supporting Information with a file name “PRISMA checklist".

Additional Editor Comments:

Dear Mr Aklilu Habte,

Academic editors’ comments

The topic of the manuscript is interesting. Nevertheless, the reviewers raised several concerns: considering this point, I invite authors to perform the required major revisions.

# You should modify the title “Determinants of respectful maternity care during childbirth in Ethiopia: A systematic review and Meta-analysis”

You should give line numbers across the manuscript

#Abstract

The background is too long, make shortened.

1. The first and second sentences, it is difficult to understand. Write clearly and understandable way.

2. Methods from when to June 2022?

3. In the abstract abbreviation does not recommend (CRC and HCPs).

4. As per this meta-analysis, the level of RMC in Ethiopia was low (48.44 percent), suggesting that more emphasis is needed to plan and implement intervention measures. This sentence needs modification and avoid the word level. What is your ground to say low?

5. You should forward your recommendation based on your pertinent findings.

#Introduction

1. The introduction is too long, you should be focused and addressed your objectives, what is known and what is not unknown. This article is similar to your manuscript https://pubmed.ncbi.nlm.nih.gov/30760318/.

#Methods

1. Population: Women in the reproductive age group (15-49)

Your population is childbirth, not reproductive age

2. study settings are either facility based on community-based

Result

1. Why do you exclude qualitative studies? Why not synthesise evidence from these studies?

Discussion

It is too long .

You should discuss your pertinent findings, how and why this result comes, and the limitations.

Reviewers' comments:

Reviewer's Responses to Questions

**Comments to the Author**

1. Is the manuscript technically sound, and do the data support the conclusions?

Reviewer #1: Yes

Reviewer #2: Yes

Reviewer #3: Yes

Reviewer #4: Yes

2. Has the statistical analysis been performed appropriately and rigorously? 

Reviewer #1: Yes

Reviewer #2: Yes

Reviewer #3: Yes

Reviewer #4: Yes

3. Have the authors made all data underlying the findings in their manuscript fully available?

Reviewer #1: Yes

Reviewer #2: Yes

Reviewer #3: No

Reviewer #4: Yes

4. Is the manuscript presented in an intelligible fashion and written in standard English?

Reviewer #1: Yes

Reviewer #2: Yes

Reviewer #3: Yes

Reviewer #4: Yes

5. Review Comments to the Author

Reviewer #1: This is an important and methodologically sound article, which will be of great interest and practical utility to a local audience as well as a global one. Nevertheless, there are some minor revisions that are necessary to clarify significant ambiguities stemming mostly from language issues, and the paper would be strengthened overall by good copyediting to improve the quality of the writing and thus the clarity of the content.

Please find my inputs below:

Abstract

In the abstract (but not the body of the paper) there is a typographical error: “DerSimonian Laired”.

CRC: this abbreviation appears throughout the paper, in reference to some aspect of the Ethiopian government’s Health Sector Transformation Plan that emphasizes “compassionate, respectful care”. It is unclear if this is a specific training program with a defined curriculum, learning objectives and outcome measures, such that it could be replicated with similar results. These details are important as it emerges as a significant variable in the logistical regression reported. If so, the program should be described and cited; from a language perspective it should appear with first letters capitalized. If not, I would suggest citing the HSTP, and adding some discussion and calls for future research to identify what elements of that program are effective in strengthening RMC.

Introduction

What is Reference 3 and how is it relevant to the point? There are two rights-based frameworks, White Ribbon and Khosla et al. that I suggest should be cited here.

Maternity care includes more than monitoring. See ILO ISCO-8 classification of occupations for midwifery professionals for a concise, yet comprehensive list of responsibilities.

Re. the following sentence, “Although several epidemiological studies on the magnitude and determinants of RMC in Ethiopia have been conducted, the results have been inconsistent and varied,”

1) I am not aware of studies that expressly measure the prevalence of RMC; many measure the prevalence of Disrespect and Abuse (DA)/mistreatment. Does this study derive the prevalence of RMC from studies that aim to measure DA, and if so by what methodology? Or are these all studies that specifically measure RMC, and if so, how was RMC defined and operationalized in these original studies? Was it the same?

2) I suggest reading and referencing here the study by Sando et al.: https://reproductive-health-journal.biomedcentral.com/articles/10.1186/s12978-017-0389-z

Methods, Measurement of the outcome of interest

Similar comment to the above: There are so few studies that measure RMC and there have been no standard definitions for RMC to my knowledge other than this paper (https://pubmed.ncbi.nlm.nih.gov/34598705/) that I am concerned whether the studies truly measured prevalence of RMC or whether they looked rather at DA/mistreatment and extrapolated RMC. RMC cannot be construed as the simple absence of DA, although this becomes nuanced. It would be important to understand how the authors calculated the prevalence of RMC in the base studies.

Re. the following sentence, “RMC is a universal human right that must be provided to all childbearing women in every health system and is measured by four performance standards (friendly care, timely care, discrimination-free care, and abuse-free care),” What is the citation for this? Or was this the study definition? If so, it should be explicitly stated and operational definitions provided.

Characteristics of Included Studies

Re. the sentence, “All of the included studies collected data through face-to-face interviews with a pre

tested, interviewer-administered questionnaire,” in what setting? In what timeframe relative to birth? See Sando et al. for comparison of prevalence by data collection setting and timing.

Determinants of RMC

In this section, there are some language issues that obscure the meaning of the results presented.

Re. the following sentence, “As significant determinants of RMC, six variables were identified: giving birth during the day, planning status of previous pregnancy, having ANC visit, experiencing an

obstetric complication, and receiving service from health care providers who were trained on

CRC”:

1) Does this refer to the current pregnancy, or a previous pregnancy resulting in a previous birth? I think it means the current pregnancy preceding the current birth. Please clarify.

2) Does this refer to any ANC or adequate ANC? I believe from reading the whole paper it refers to adequate ANC, but this should be clarified. These are significantly different.

3) Again, if CRC is a significant determinant, it should be defined and cited at first mention in this paper and its essential elements described.

4) Most importantly, the directionality of the association between obstetric complications and RMC is NEGATIVE. This MUST be clarified. The way that these four significant factors are lumped together and jointly described as determinants of RMC is very misleading and confusing. The study results demonstrate that obstetric complications are a determinant of reduced odds of RMC, the opposite of the three other variables. This distinction should be made very clear every time it is mentioned.

Page 13: Same comment:

The fact that the association between RMC and obstetric complication is not conveyed clearly in the summary descriptions above and this is a very important distinction from the other factors that are positively associated. This must be made explicit in each instance it is mentioned.

Discussion

Page 15: Re. this sentence, “As per subgroup analysis results, the highest and the lowest level of RMC was reported in the Oromia, 58.01%(95% CI: 42.44, 73.58) and Benishangul Gumuz region 12.65% (95% CI: 9.45, 15.90) respectively, while it was lowest in.” Correct significant typos and missing words here.

Page 15: Re. this sentence, “Accordingly, receiving service from CRCtrained health care providers, having ANC visits, pregnancy planning status, giving birth during the daytime, and experiencing an obstetric complication were identified as determinants of RMC” : Again, this is incorrect and misleading. OB complication is NOT a determinant of RMC but a barrier to RMC or risk factor for low RMC.

Page 16: Re. the following sentence, “The possible justification could be that women who had adequate ANC visits had a better chance of acclimating to the health facility setup and developing close relations with the HCP. All of these are essential in ingraining trust in the facility's services, which resulted in RMC[39, 53]” please see my comment:

Since provider behaviors are the basis of RMC, an explanation that centers the change in provider behavior toward clients who had adequate ANC, or a change in the dynamic between provider and client might be mentioned here. Otherwise, is the hypothesis that women's attitudes or perceptions changed if they had adequate ANC? The WHO Bulletin definitions of DA by Freedman et al might be interesting to consult here (https://apps.who.int/iris/handle/10665/271621).

Page 16: Re. the following sentence, “Rather than mistreating women who have had an unplanned pregnancy, it would be recommendable to focus on preventing those very pregnancies through the provision of contraception,”: A more woman-centered way to express this might be, "assisting women to meet their need for contraception"...

Page 17: Re. “This could be because women who experienced complications during labor are more

likely to develop postpartum blues and depression, which can impede and lower the process

and level of receiving RMC.” This statement needs a reference citation.

Re. “In addition, complicated labor necessitates frequent and meticulous follow-up, which exhausts the provider and may result in service abandonment,” see my comment: There are a number of dimensions of DA that could be associated with complicated birth theoretically (e.g., unconsented care, lack of information and choice, lack of respect for values and preferences, exclusion of companion of choice, lack of privacy...) therefore, this merits further discussion and literature search/citations.

Reviewer #2: Dear PLOSE One team of editorials, thank you for giving me the chance to review the manuscript entitled "Towards the quality of maternal and newborn health care: The level and determinants of respectful maternity care during childbirth in Ethiopia: A systematic review and meta-analysis".

Reviewer Comments to the Author

This study gives very important results regarding the level and determinants of respectful maternity care during childbirth. However, in a few areas, here are my comments.

General Comments

Why do you review the articles on RMC only from Ethiopia?

The abstract is many worded, Abbreviations are used in the abstract section. need correction

What is the unique characteristic of the quality of maternal and newborn health care? What is the level of quality care? The determinants of respectful maternity care are not mentioned clearly in the introduction section as mentioned in the results. (e.g., friendly care, timely care, discrimination-free care, and abuse-free care).

Make sure that all the elements of the background section are fulfilled. Describe in a sequential way what is known and unknown and what gaps you want to fill with your study.

The introduction, results, and conclusion should be in line with the research objectives.

The topic can be refined ( or make short)

I think the population will be pregnant women. Reproductive age group is a vague term for RMC.

The inclusion of data from unpublished studies can itself introduce bias.

Insufficient citation, particularly in discussion for safe interpretation.

Use correct tense, grammar, sentence, spelling, paraphrase, consistency…etc needs correction

.

Reviewer #3: Dear Editor/ authors

Despite writing nicely I felt some issues in the manuscript. I suggest addressing these issues to accept for publishing it, My suggestions/ comments are as follows.

Abstract or summary section RMC measurement method is not clearly defined with specifying measurement scale (count, ordinal continuous, or binary).  The study claimed the use of the random effect model to analyze 43 (some places 38) studies.  Studies use a random effect model in panel data. This study evaluated mostly the results of cross section studies. The cluster variable (whether year, region, or something else) of this study is not clear.  If the studies were clustered, was that sample enough or statistical analysis? Missing full form of AOR. 

Introductions section:

The writing of the introduction section is too long but it missed explaining vital things.  In the last paragraph of the introduction section, the current knowledge of RMC requires further elaboration to justify. The message of the statement "the results have been inconsistent and varied"  is inadequate.  

Method section

An illustration of the data screening process in the figure would make the paper more appealing to readers. Please refer to other meta-analysis-based papers in the health sector. 

Results 

I suggest placing most figures and tables in the main body. Readers find it difficult to follow materials in supplements and appendix. The determinant variables are a vital part of this study. Presenting the variables in the main body instead of S4 file would increase the values of this paper.  

Discussion: This study benchmarked with meta-studies of different countries. I am doubtful whether the dates of the publications are of similar times. 

Conclusion section: Hardly a few findings are generalized in the conclusion section. Most of the space is used for recommendations.  I would avoid the strong word "should" to write recommendations.

Reviewer #4: The systematic review and metaanalysis on the determinants of RMC was good. It would be worthwhile for policymakers to plan and improve childbirth care. The following changes are required in this manuscript:

1. Abstract: Go through lines 4-7 and it's better to remove from the abstract and include in the introduction section.

- used the terms "prevalence" or "incidence" of RMC instead of "level of RMC"

- Remove the last two lines of conclusion in the abstract: " A due empahsis.

2. Introduction:

The introduction is too long, make it clear and to the point, focusing on research questions. I suggested including studies related to variables that affect RMC, the prevalence of RMC in ethopis health facilities or community-based facilities, and any differences in the prevalence of RMC in different sectors.

The existing evidence of mistreatment and abuse or other components of RMC

Then fill in the gaps with the study in the last paragraph.

3. Study selection process: include how many articles for SR and metaanalysis.

4. Incusion criteria: study design: make clear regarding reporting the level of RMC, I suggested to replace the level of RMC throughout the study by Prevalence.

5. Could you explain which threshold of p value has been used for statistical significance when using the Cochrane Q test to determine statistical heterogeneity?

6. Results

Table 1 suggest to write Prevalence of RMC in heading

Table 2: wirite components of RMC instead of domains of RMC. explain its details in the results section.

Explain sensitivity analysis based on..(Fig 6); elaborate these information in the result section.

7. Discussion:

-The discussion was so long. Could you please focus on the main objective of the study? Look for the first paragraph of the conversation (you can make it very brief).

-Discussed regarding components of RMC.

-Could you include some other critical factors that influence components of RMC or the overall prevalence of RMC? 

Include the current study's strength in the last paragraph before the limitation.

_Provide references in discussions ection line start...On the other hand, the tendency to receive

low RMC during the night shift may be explained by the low staff number -to- obstetric cases

-Provide reference for line start..... In addition, complicated labor necessitates frequent and

meticulous follow-up, which exhausts the provider and may result in service abandonment

- Check reasons for this and reference (this might not be the case during child birth)..line start from...This could be because women who experienced complications during labor are more

likely to develop postpartum blues and depression, which can impede and lower the process

and level of receiving RMC.

8. Conclusion: Include some information regarding the strength of evidence and write some of the geographical differences in RMC prevalence in Ethiopia. You already recommended removing the duplication of information in the discussion section. focus on the main findings of the study.

6. PLOS authors have the option to publish the peer review history of their article (what does this mean?). If published, this will include your full peer review and any attached files.

Reviewer #1: No

Reviewer #2: No

Reviewer #3: No

Reviewer #4: **Yes: **Rojana Dhakal, School of Health and Allied Sciences, Pokhara University

---

## [Author Response · Author response to Decision Letter 0]

5 Oct 2022

Attached as "Response to reviewers" in the submission system.

---

## [Decision Letter · Decision Letter 1]

31 Oct 2022

PONE-D-22-19516R1The prevalence of respectful maternity care during childbirth and its determinants in Ethiopia: A contemporaneous systematic review and Meta-analysisPLOS ONE

Dear Dr. Habte Hailegebireal,

Thank you for submitting your manuscript to PLOS ONE. After careful consideration, we feel that it has merit but does not fully meet PLOS ONE’s publication criteria as it currently stands. Therefore, we invite you to submit a revised version of the manuscript that addresses the points raised during the review process.

We look forward to receiving your revised manuscript.

Kind regards,

Zemenu Yohannes Kassa, Msc

Academic Editor

PLOS ONE

Journal Requirements:

Additional Editor Comments:

Dear Dr Hailegebireal,

Thank you for considering PLOS ONE.

I assessed your revision as Editor I am pleased to inform you that it is potentially publishable in PLOS ONE and I invite authors to perform the required minor revisions.

in the title, you should remove contemporaneous . you should revise the grammar.

line 49 rewrite again,line 287,288,325,337, 366, 384, 460 and 470 grammar error.

Reviewers' comments:

Reviewer's Responses to Questions

**Comments to the Author**

1. If the authors have adequately addressed your comments raised in a previous round of review and you feel that this manuscript is now acceptable for publication, you may indicate that here to bypass the “Comments to the Author” section, enter your conflict of interest statement in the “Confidential to Editor” section, and submit your "Accept" recommendation.

Reviewer #3: (No Response)

Reviewer #4: All comments have been addressed

2. Is the manuscript technically sound, and do the data support the conclusions?

Reviewer #3: Yes

Reviewer #4: Yes

3. Has the statistical analysis been performed appropriately and rigorously? 

Reviewer #3: Yes

Reviewer #4: Yes

4. Have the authors made all data underlying the findings in their manuscript fully available?

Reviewer #3: No

Reviewer #4: Yes

5. Is the manuscript presented in an intelligible fashion and written in standard English?

Reviewer #3: Yes

Reviewer #4: Yes

6. Review Comments to the Author

Reviewer #3: (No Response)

Reviewer #4: The review was good and highlights the major components of respectful maternity care in Ethiopia. I suggest checking for spelling errors in manuscripts, tables, and figures.

7. PLOS authors have the option to publish the peer review history of their article (what does this mean?). If published, this will include your full peer review and any attached files.

Reviewer #3: No

Reviewer #4: **Yes: **Rojana Dhakal

---

## [Author Response · Author response to Decision Letter 1]

31 Oct 2022

General comment and suggestion

I assessed your revision as Editor I am pleased to inform you that it is potentially publishable in PLOS ONE and I invite authors to perform the required minor revisions.

Response: I thank you for the time and effort made to review the manuscript in detail, for your constructive comments, and for the opportunity to revise and resubmit. After completion of the suggested edits, the revised manuscript has benefitted from an improvement in the overall presentation and clarity. We have highlighted the document to indicate any changes in words, phrases, or sentences.

Comment 1: In the title, you should remove contemporaneous.

Response: We have amended the title as per your suggestion and highlighted it on the “title page” of the “Revised manuscript with track changes” Line 2-3, Page1. 

Comment 2: line 49 rewrite again, with lines 287,288,325,337, 366, 384, 460 and 470 grammar error.

Response: thank you for your comment. As per your suggestion, we have tried to correct those sentences in the above-mentioned lines. All the corrected versions were highlighted throughout the revised version of the manuscript.

---

## [Editor Report · Decision Letter 2]

6 Nov 2022

The prevalence of respectful maternity care during childbirth and its determinants in Ethiopia: A  systematic review and Meta-analysis

PONE-D-22-19516R2

Dear Dr. Hailegebireal,

We’re pleased to inform you that your manuscript has been judged scientifically suitable for publication and will be formally accepted for publication once it meets all outstanding technical requirements.

Kind regards,

Zemenu Yohannes Kassa, Msc

Academic Editor

PLOS ONE
---

## [Editor Report · Acceptance letter]

9 Nov 2022

PONE-D-22-19516R2 

The prevalence of respectful maternity care during childbirth and its determinants in Ethiopia: A systematic review and Meta-analysis 

Dear Dr. Habte Hailegebireal:

I'm pleased to inform you that your manuscript has been deemed suitable for publication in PLOS ONE. Congratulations! Your manuscript is now with our production department. 

Kind regards, 

on behalf of

Dr. Zemenu Yohannes Kassa 

Academic Editor

PLOS ONE